# Revisiting Static Feature-Based Android Malware Detection using Machine Learning

## Abstract

Machine learning models are widely used for malware detection based on static features. However, many studies in this area show inconsistencies in their experimental settings, often failing to adequately consider the nature of the datasets, the underlying tasks, and the models being evaluated. This lack of standardization complicates the reproducibility of results on public datasets. In this paper, we address these challenges by proposing a more rigorous experimental and model selection methodology for malware detection. Specifically, we focus on Android malware detection using two public datasets evaluated under offline and continuous active learning settings. We implement six machine learning models of varying complexity across diverse experimental configurations. Our results show that tree-based methods, such as XGBoost, frequently outperform advanced neural networks in various scenarios. To promote reproducibility, we open-source our code, ensuring it is extensible for incorporating new models and datasets.

## 1 Introduction

Machine learning (ML) has become indispensable in computer security and is extensively used in literature for malware detection (Arp et al., 2014; McLaughlin et al., 2017; Mariconti et al., 2017; Gopinath & Sethuraman, 2023). These models learn patterns from data, adapt to new threats, and scale far beyond the capabilities of traditional signature-based systems. However, despite their potential, the design and deployment of ML systems in security applications face significant challenges (Arp et al., 2022). Key issues, as observed in prior works (Arp et al., 2022; Flood et al., 2024; Verma et al., 2019), include sampling bias, incorrect labels, data snooping, inappropriate threat models, and unsuitable baselines. Additionally, reproducibility studies of these methods are often neglected, leading to claims that cannot be reliably corroborated. This lack of rigor in the design and evaluation process undermines the validity of many reported findings due to poor reproducibility (Olszewski et al., 2023). Addressing these challenges requires a focus on thorough evaluation practices, ensuring that results are consistently reproducible across different environments and datasets.

An example of reproducibility challenges is illustrated in Figure 1, where we examine the reproducibility of results reported by Chen et al. (2023a) for Android malware detection using continuous active learning. We reproduced the experiments using the artifacts provided by the authors, including code and datasets. By using the provided best set of hyperparameters for the Drebin-based dataset (Arp et al., 2014), we conducted the experiments with five different random seeds. We then compared the best and worst-performing models on the test set. The best model achieved an average F1 score of 79.26% over the test months, while the worst model scored 68.65%, indicating a performance difference of 10.6% solely due to variations in random seed initialization. This example highlights how reliably reproducing prior results in Android malware detection can be fraught with challenges, which, if not adequately addressed, can hinder progress in this research field.

In this paper, we explore the specific challenges associated with reproducibility and replicability in Android malware analysis. By examining key aspects of the dataset and methodology, we shed light on the factors that hinder the reproducibility of research outcomes in this domain. We identify issues such as data duplicates, inadequate model selection, hyperparameter tuning, and flawed evaluation strategies. These contribute to disparities in results obtained under the same experimental settings, leading to variability in model perfor-

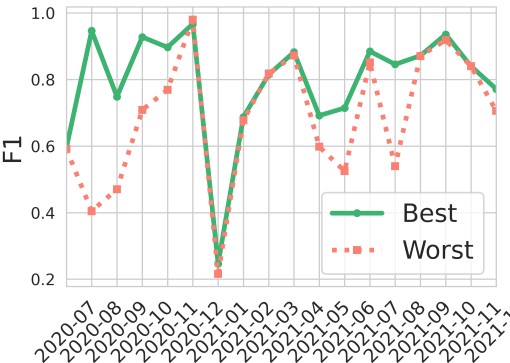 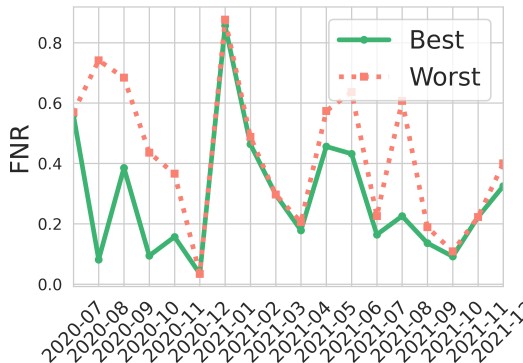

Figure 1: Comparison of best and worst performing neural networks in Android malware detection using state-of-the-art continuous active learning method from Chen et al. (2023a). Models were initialized with 5 different random seeds. The average F1-score over months differs by 10.6%, and the False Negative Rate (FNR) differs by 13.9% between the two models, despite using the same hyperparameters. This highlights reproducibility challenges in machine learning research for Android malware detection.

mance and thus hurting reproducibility. We systematically discuss how these factors cause irreproducibility and propose ways to mitigate them.

Our study primarily focuses on reproducing the results of a state-of-the-art method for continuous active learning in Android malware detection (Chen et al., 2023a). However, we expand the scope by considering more diverse experimental settings and addressing general reproducibility issues in Android malware detection. We utilize six different models: Random Forest (Breiman, 2001), Support Vector Machine (Arp et al., 2014), eXtreme Gradient Boosting (Chen & Guestrin, 2016), Multilayer Perceptron (Rumelhart et al., 1986), Supervised Contrastive Classifier (Yang et al., 2021), and Hierarchical Contrastive Classifier (Chen et al., 2023a), previously used in Android malware analysis. We analyze both offline learning, where the model is trained once, and continuous active learning, where the model is periodically retrained with a subset of annotated samples. We perform detailed model selection and hyperparameter tuning for all models in both settings. Our analysis shows how data duplicates affect reproducibility, how adequate hyperparameter tuning of baseline models can yield surprising results compared to more complex models, and offers recommendations for enhancing reproducibility in future research.

**Key Contributions**

1. We identify and discuss key issues and pitfalls in datasets and methodologies in Android malware analysis that lead to poor reproducibility of previously reported results. We propose solutions to address these pitfalls, aiming to improve reproducibility and enable fairer comparisons across different models.

2. We implement state-of-the-art machine learning methods for Android malware detection in both offline and continuous active learning settings. Our rigorous study of model performance across diverse experimental settings reveals that simpler baseline methods can often outperform more complex models when properly tuned.

3. We open-source our code[1] (to be made public upon publication) for the wider community to facilitate malware analysis. Our code is easily extensible for new models and tasks, supporting future research on Android malware detection to improve the reproducibility of published results.

---

[1] https://anonymous.4open.science/r/maldetect2025

## 2 Background & Related Work

### 2.1 Reproducibility & Replicability

We adhere to ACM's definitions of reproducibility and replicability, standardized in 2020.[2] Computational reproducibility refers to the ability of an independent team to achieve a study's results using the original study's artifacts (Gundersen & Kjensmo, 2018). This ensures that the reported findings are valid under specified conditions. Olszewski et al. (2023) conducted a longitudinal study on the reproducibility of security papers published in Tier 1 venues (2013-2022), identifying challenges in reproducing results with author-provided artifacts and proposing mitigation strategies.

Replicability, in contrast, implies that an independent group can obtain the same results using artifacts they develop independently. Replicability studies generally involve different datasets and aim to confirm previous research results, considering the underlying system's inherent uncertainty.

Our study examines both aspects of machine learning research for Android malware detection. For instance, the analysis shown in Figure 1 underscores reproducibility issues in prior research, even when author-provided artifacts are used, due to unaccounted variability in published results. Additionally, we discuss pitfalls that must be addressed for a study's claims to be replicable with independently developed artifacts. In this paper, we use the term reproducibility to broadly refer to both computational reproducibility and replicability.

### 2.2 Android Malware Detection

Android malware detection using machine learning classifies applications as benign or malicious (Liu et al., 2020). Three main types of features are used: static, dynamic, and hybrid. Static features are obtained by analyzing the app's source code or related information (Arp et al., 2014; Mariconti et al., 2017; Zhang et al., 2020). Specifically, the primary focus for Android applications is the APK file, the installation package, which includes components like AndroidManifest.xml and smali files obtained through decompilation. Dynamic features are acquired by observing the app's behavior in real or emulated environments, such as sandboxes (Jannat et al., 2019; Shankar et al., 2017; Zhang et al., 2018). Hybrid features combine both static and dynamic characteristics (Choudhary & Kishore, 2018; Martinelli et al., 2017).

Extracted features are input into machine learning models like Random Forests (Breiman, 2001), Support Vector Machines (Arp et al., 2014), and Multilayer Perceptrons (Rumelhart et al., 1986) to classify apps as malware or benign. The choice of features and models depends on detection objectives and resources. Static analysis is fast and suitable for large-scale detection, whereas dynamic analysis can be more accurate but resource-intensive (Aghakhani et al., 2020). This work focuses on static-feature-based machine learning methods due to their prevalence in the literature (Jordaney et al., 2017; Pendlebury et al., 2019; Zhang et al., 2020; Barbero et al., 2022; Chen et al., 2023a).

**Concept Drift:** Malware detection systems encounter challenges due to the evolving nature of malware, leading to concept drift. This drift can arise from new malware families, behavioral changes, evasion attempts, or updates in API semantics. Traditional supervised learning models trained on static datasets may become outdated as new malware types emerge, reducing detection accuracy (Yang et al., 2021; Chow et al., 2023; Pendlebury et al., 2019; Xu et al., 2019). These models often fail to recognize emerging variants, resulting in higher false negatives. To address this, continuous learning methods can enable models to adapt to new data continuously and stay current with malware trends (Jordaney et al., 2017; Chen et al., 2023a).

**Reproducibility of Research in Android Malware Detection Using ML:** Previous studies have assessed issues in reported results for Android malware detection (Zhao et al., 2021; Irolla & Dey, 2018), attempted to replicate prior research (Daoudi et al., 2021), or addressed limitations in practical environments (Gao et al., 2024). However, these studies do not consider the continuous learning setup, often rely on single global metrics like AUC or F1-score, lacking nuanced analysis of monthly performance variations, factors influencing reproducibility, and inherent variances in machine learning models due to their stochastic

---

[2]https://www.acm.org/publications/policies/artifact-review-and-badging-current

| Dataset | Split | Duration | Benign Apps | Malicious Apps | Total | Malware Families | Feature Dimension |
|---------|-------|----------|-------------|----------------|-------|------------------|-------------------|
| Drebin | Train | 2019-01 to 2019-12 | 40,947 | 4,542 | 45,489 | 121 | |
| | Validation | 2020-01 to 2020-06 | 18,109 | 2,028 | 20,137 | 67 | 16,978 |
| | Test | 2020-07 to 2021-12 | 30,797 | 3,631 | 34,428 | 71 | |
| APIGraph | Train | 2012-01 to 2012-12 | 27,472 | 3,061 | 30,533 | 104 | |
| | Validation | 2013-01 to 2013-06 | 21,310 | 2,366 | 23,676 | 115 | 1,159 |
| | Test | 2013-07 to 2018-12 | 240,729 | 25,377 | 266,106 | 418 | |

Table 1: Summary statistics of the datasets used in the study

nature. Our work aims to address these gaps by analyzing conditions that can lead to poor reproducibility in different experimental settings and how to mitigate them.

## 3 Study settings

### 3.1 Machine Learning

In this study, we consider two distinct machine-learning settings:

1. **Offline Learning:** In this classical supervised setting, the model is trained once on the training dataset and evaluated on the test set. Offline learning for malware detection involves using annotated data from a specific period (e.g., one year) to train a model and evaluate its performance on future unseen samples.

2. **Continuous Active Learning:** While offline settings may suffice when the data distribution remains constant, malware detection is susceptible to concept drift. Continuous active learning addresses concept drift by periodically retraining the model (Jordaney et al., 2017; Zhang et al., 2020; Chen et al., 2023a). This cost-effective approach focuses on annotating only the samples where the model is least confident, saving annotation resources. Although determining when to retrain remains challenging, a common practice is to retrain the model monthly. We adopt this approach in our study.

We treat these settings separately because each requires different model selection methods, such as hyperparameter tuning, as noted in prior work (Chen et al., 2023a).

### 3.2 Datasets

We utilize the publicly available dataset from Chen et al. (2023a), which includes two datasets based on static feature types: Drebin (Arp et al., 2014) and APIGraph (Zhang et al., 2020). Hereafter, we refer to these as the Drebin and APIGraph datasets. Drebin uses eight different sets of features, including access to hardware components, requested permissions, app components, intents, usage of restricted API calls, used permissions, suspicious API calls, and network addresses. APIGraph employs API semantics to cluster similar APIs, significantly reducing the feature space. The resulting feature set is binary for both datasets, indicating whether a feature is present or absent for a specific sample.

The Drebin dataset contains applications collected from 2019 to 2021, while the APIGraph dataset includes applications from 2012 to 2018. Both datasets address temporal and spatial biases present in malware datasets (Arp et al., 2022; Pendlebury et al., 2019). They consist of approximately 90% benign and 10% malware applications collected from each month, reflecting their real-world ratios. Samples are ordered and approximately evenly distributed over their respective periods. Table 1 provides a summary of the datasets.

Following Chen et al. (2023a), we use data from the first year as the training set, the next six months for validation, and the remaining months for testing. For offline learning, only the training set data is used to train the model. In the continuous active learning setup, this data is used to train the initial model, which is then periodically retrained using data from subsequent months from the validation and test set. The validation set is used to tune the model hyperparameters in both experimental settings.

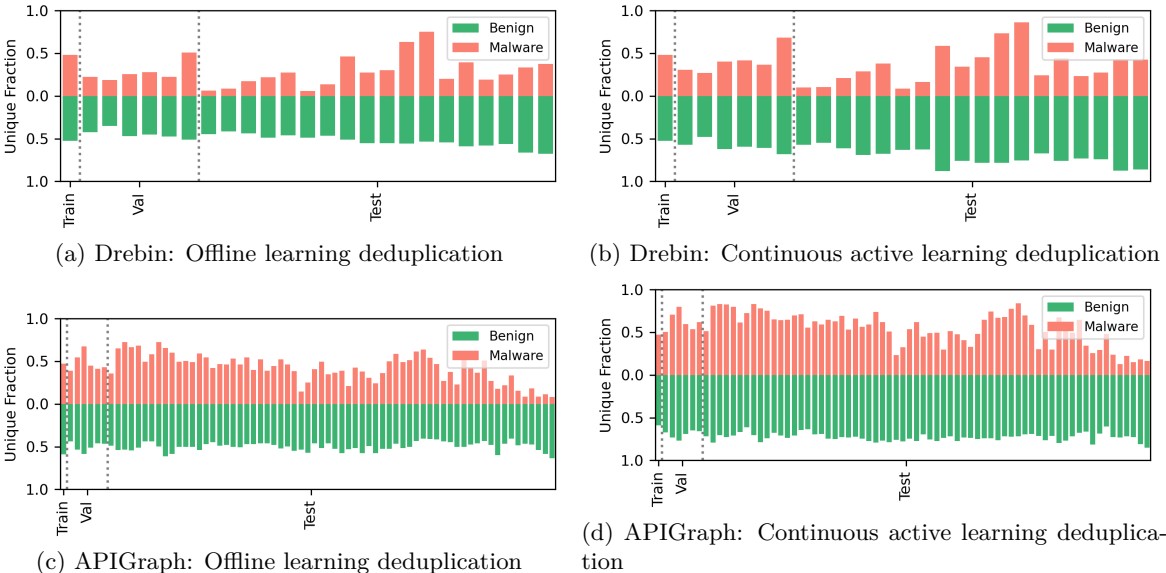

(a) Drebin: Offline learning deduplication

(b) Drebin: Continuous active learning deduplication

(c) APIGraph: Offline learning deduplication

(d) APIGraph: Continuous active learning deduplication

Figure 2: Fraction of unique samples retained after the deduplication process

## 4 Pitfalls

In this section, we examine the challenges in the existing literature on Android malware detection using static features. We focus on dataset and methodological issues that impact the reproducibility of reported results.

### 4.1 Data Duplicates

Issues of duplicates in Android malware datasets have been reported in prior studies (Irolla & Dey, 2018; Zhao et al., 2021). The study by Irolla & Dey (2018) analyzed duplicates in the original Drebin dataset (Arp et al., 2014) using opcode sequences and found that 50.65% of malware samples were unique, with the rest being duplicates. They demonstrated that removing duplicates can alter the performance rankings of different machine learning models. Zhao et al. (2021) examined duplicates in four malware datasets using APK, DEX, opcode sequences, and API calls to identify them. They also evaluated the impact of duplicates on multiple machine-learning models.

Our study differs from prior works in key ways. First, we are the first to address the reproducibility of reported results in this context. Second, we provide a time-aware analysis of duplicates' effects on malware detection models, as global metrics (e.g., F1-score) may miss monthly variations (Pendlebury et al., 2019). Lastly, unlike prior studies focused only on offline learning, we assess duplicates' impact in offline and continuous learning settings, which require different deduplication approaches, as discussed later.

#### 4.1.1 Defining Duplicates

We define two samples as duplicates if their feature sets are identical after the feature extraction process, meaning they are presented as identical inputs to the machine learning model. Specifically, two samples $X$ and $Z$ are considered duplicates if $x_1 = z_1, x_2 = z_2, \ldots, x_n = z_n$ for an $n$-dimensional feature vector:

$$X = (x_1, x_2, \ldots, x_n), \quad Z = (z_1, z_2, \ldots, z_n)$$

This definition ensures that duplicate detection is based entirely on the processed feature representations, independent of how the features are derived. This approach contrasts with prior studies (Irolla & Dey, 2018; Zhao et al., 2021), which define duplicates based on raw representations such as opcode sequences. By focusing on feature-space duplicates, we aim to analyze how duplicates influence the reproducibility of

results for machine learning models trained on identical feature sets. Notably, despite using this feature-space definition, we observe a similar percentage of duplicates in the datasets as reported in prior studies.

It is important to highlight that detecting accurate duplicates is particularly challenging in malware analysis, which we further discuss in Section 8.

### 4.1.2 Deduplication

The process of *deduplication* ensures that the dataset contains unique samples. We define two types of deduplication based on different experimental settings in our study:

**Offline Learning Deduplication:** In offline learning, the model is trained once on a training set and evaluated on validation and test sets. Deduplication involves selecting unique samples within and across data splits. Since the model is evaluated monthly, two levels of deduplication are required. First, we remove samples that appear in earlier data splits: duplicates in the validation set are removed if present in the training set, while the test set excludes duplicates from both the training and validation sets. Second, within each split, only the earliest occurrence of a sample is retained based on its appearance time. This process ensures that all samples across the training, validation, and test sets are unique while preserving the earliest unique samples.

**Continuous Active Learning Deduplication:** In continuous active learning, the definition of duplicates evolves due to monthly model retraining and changes in behavior, such as catastrophic forgetting (Rahman et al., 2022). Duplicate samples are first removed from the training split, following the same approach as offline learning. For the validation and test sets, duplicates are removed only within the same month, not across different months. This allows duplicates to appear in multiple months of the test set while ensuring that each month contains unique instances. This approach is crucial for continuous learning, as periodic retraining requires the model to recognize recurring samples accurately.

### 4.1.3 Duplicate Statistics in the Datasets

Figure 2 illustrates the proportion of unique samples in the datasets after the deduplication process in both experimental settings. Ideally, this fraction would be 1 for both malware and benign apps, indicating no duplicates. However, our analysis shows that many duplicates are present, with varying degrees across different splits. Notably, around 50% of the training set for the Drebin and APIGraph datasets consists of duplicate samples in benign and malware categories.

We observe two key patterns regarding duplicates in the validation and test sets. First, the fraction of duplicate samples varies monthly. Second, malware samples tend to contain more duplicates than benign samples, with certain months particularly affected. For instance, in the first test month of the Drebin dataset (July 2020), only 6.30% of malware samples are unique in the offline-learning setting, and 9.96% are unique in the continuous-learning setting.

### 4.2 Model Selection Strategy

A common approach for evaluating machine learning models involves splitting the dataset into training, validation, and test sets. The validation set is essential for model selection and hyperparameter tuning to prevent biased parameter choices (Arp et al., 2022). However, previous research on Android malware detection often omits a separate validation set, leading to incomplete evaluations (Arp et al., 2022). Our study adopts the same train, validation, and test split as in Chen et al. (2023a) for offline and continuous learning settings.

Another common issue is the inadequate tuning of baseline methods (Arp et al., 2022). This often results in insufficient hyperparameter tuning for simpler models compared to more complex ones. For example, prior research has shown that appropriately tuned tree-based methods frequently outperform deep neural networks on tabular datasets (Grinsztajn et al., 2022; McElfresh et al., 2024), even though neural networks excel in tasks like computer vision and natural language processing (Krizhevsky et al., 2012; He et al., 2016; Vaswani et al., 2017; Devlin et al., 2019).

In Android malware detection, various models have been used in literature, such as support vector machines (SVM), random forests, gradient-boosted decision trees, and deep neural networks (Arp et al., 2014; Mariconti et al., 2017; Pendlebury et al., 2019; Zhang et al., 2020; Chen et al., 2023a). However, these models might not have been adequately calibrated due to the absence of a validation set or insufficient hyperparameter searches. Our analysis emphasizes extensive hyperparameter searches across different baseline methods, ensuring the same hyperparameter budget is allocated to each model. This approach enables a fair comparison between models and aligns with the established norms in the domain generalization literature in machine learning (Gulrajani & Lopez-Paz, 2021).

### 4.3  Accounting for Variance in Performance

Some machine learning models, particularly neural networks, exhibit additional stochasticity independent of their hyperparameters. Their sensitivity to initial random weight initialization can lead to variations in results that depend on the training system rather than the model parameters (Bouthillier et al., 2021; Picard, 2021). It is common practice in the deep learning community to evaluate performance across multiple random seeds on the test set and report the average (Bouthillier & Varoquaux, 2020) to address this. However, prior studies on Android malware detection often neglect this variation, leading to irreproducible outcomes when experiments are conducted with different random seeds. We explore this issue in detail in Section 6.2.

### 4.4  Delayed Evaluation of Models

Splitting the dataset into non-overlapping temporal train, validation, and test sets facilitates better model selection. However, this approach introduces a delay between the training and test datasets. While this delay is not problematic for classical supervised learning, which assumes an independent and identically distributed setting, it poses challenges for malware detection. For example, in our study and in Chen et al. (2023a), there is a six-month gap between the training and test datasets. Directly evaluating the test set using a model trained solely on the training data may not accurately reflect practical deployment scenarios. Ideally, the model should be trained with more recent data before deployment (Arp et al., 2022).

To address this, we merge the training and validation datasets after hyperparameter tuning and evaluate the model on the test dataset. We propose that this approach provides a fairer estimate of model performance in offline learning, where the model is inherently delayed by the time the test month begins. Merging validation data with training data is a common practice in temporal domain generalization (Yao et al., 2022). We refer to this as the **Merged Training** setup, in contrast to the **Holdout Training** setup, where the model is trained only on the training dataset and evaluated on the test dataset. The impact of these two evaluation settings is demonstrated in Sections 6 and 7.

## 5  Experiments

### 5.1  Machine Learning Models

We use six models commonly employed in prior works on Android malware detection: Random Forest (RF) (Breiman, 2001), Support Vector Machine (SVM) (Cortes, 1995), eXtreme Gradient Boosting (XGBoost) (Chen & Guestrin, 2016), Multilayer Perceptron (MLP) (Rumelhart et al., 1986), Supervised Contrastive Classifier (SCC) (Khosla et al., 2020), and Hierarchical Contrastive Classifier (HCC) (Chen et al., 2023a). SCC leverages supervised contrastive learning to group samples with the same label closer in the feature embedding space while pushing apart those with different labels. Chen et al. (2023a) introduced HCC, an enhanced supervised contrastive classifier with a hierarchical contrastive loss function. This approach ensures that samples from the same malware family are more similar than those from different families in the embedding space. Appendix A provides additional details about the models.

### 5.2  Active Learning Sample Selection Strategy

Active learning aims to improve model performance with minimal labeled data by selectively choosing the most informative samples from a large pool of unlabeled data (Ren et al., 2021). For all models except HCC,

we adopt the baseline uncertainty sampling strategy (Lewis & Gale, 1994). The key idea of uncertainty sampling is to select samples for which the model is least confident. For our models, this involves computing $1 - \text{probability}$ (e.g., the softmax output for neural networks) for binary classes and selecting samples with the highest uncertainty.

For HCC, we employ the pseudo-label sample selection method described in Chen et al. (2023a). This approach uses pseudo-labels as ground truth to compute the loss function. It calculates the pseudo-loss for contrastive learning by comparing the embeddings of test samples with those of nearby samples in the training dataset. The method then combines this contrastive loss with the binary cross-entropy loss, similar to the loss function used during training, to identify the most uncertain samples.

### 5.3 Hyperparameter Tuning

We perform hyperparameter tuning independently for two experimental setups: offline learning and continuous active learning.

In the offline learning setup, each model is trained once on the training dataset, with the validation set used to determine the optimal hyperparameters. We employ random search (Bergstra & Bengio, 2012) for hyperparameter tuning, allocating a fixed budget of 200 searches per model on the validation set. This ensures a fair comparison across models and accounts for performance variations due to differing hyperparameter search spaces. We also perform hyperparameter tuning separately for the duplicated and deduplicated datasets.

In the continuous active learning setup, we adopt the strategy outlined in Chen et al. (2023a). The validation data is divided into six-month intervals. We retrain the model using active learning for each interval and evaluate it in the subsequent month. Most hyperparameters, such as batch size for neural networks, remain consistent across the initial training and subsequent retraining phases (a total of five retraining phases, as no retraining is required for the final validation month). However, specific parameters, such as the number of training epochs during retraining, may vary. For continuous active learning hyperparameter tuning, we use random search with 100 searches per model and a fixed annotation budget of 50 samples per month, following Chen et al. (2023a).

Details of the hyperparameters are provided in Appendix C.

### 5.4 Evaluation Metric

After hyperparameter tuning, we evaluate the models on the test set. The continuous active learning setup includes retraining the model each test month and evaluating it the following month. We use the same performance metrics as in Chen et al. (2023a): average F1-score, False Positive Rate (FPR), and False Negative Rate (FNR). We evaluate all models on the test set using optimal hyperparameters and five different random seeds, reporting the mean and standard deviation of the metrics. It is worth noting that SVM produces deterministic results across seeds, as it uses the same training dataset without any stochastic elements.

## 6 Offline Learning Results & Analysis

### 6.1 Malware Detection Performance

We present the results of six models in Table 2 for the Drebin dataset and Table 3 for the APIGraph dataset. These results encompass the various experimental settings discussed previously. Key findings are summarized as follows:

1. **Impact of duplicates on model performance:** Deduplicated datasets generally perform worse than those with duplicates, indicating that duplicates may inflate performance metrics. For instance, in the Drebin dataset's merged training setting, the F1-score differences for the SVM and SCC models are 17.0 and 17.3, respectively, between duplicate and deduplicated datasets.

| Model | Merged Training | | | | | | Holdout Training | | | | | |
|---|---|---|---|---|---|---|---|---|---|---|---|---|
| | Duplicated | | | Deduplicated | | | Duplicated | | | Deduplicated | | |
| | F1 | FPR | FNR | F1 | FPR | FNR | F1 | FPR | FNR | F1 | FPR | FNR |
| RF | 46.7±0.99 | 0.14±0.00 | 66.8±0.74 | 41.5±0.30 | 0.08±0.00 | 72.5±0.25 | 39.4±1.31 | 0.16±0.03 | 72.4±1.03 | 33.2±1.01 | 0.14±0.04 | 78.7±0.71 |
| SVM | 63.6±0.00 | 0.80±0.00 | 47.0±0.00 | 46.6±0.00 | 0.60±0.00 | 65.6±0.00 | 47.2±0.00 | 0.92±0.00 | 63.2±0.00 | 33.8±0.00 | 0.70±0.00 | 76.7±0.00 |
| XGBoost | 63.7±2.45 | 0.18±0.01 | 48.9±2.80 | 59.5±0.46 | 0.39±0.02 | 52.1±0.59 | 45.2±1.92 | 0.24±0.04 | 67.3±2.01 | 47.3±0.78 | 0.73±0.02 | 63.4±0.66 |
| MLP | 66.2±3.69 | 0.41±0.08 | 45.8±4.10 | 53.6±2.08 | 0.39±0.08 | 59.5±2.12 | 44.3±2.07 | 0.87±0.06 | 66.7±1.72 | 40.9±1.05 | 0.57±0.04 | 69.8±0.82 |
| SCC | 74.7±0.78 | 0.53±0.09 | 35.0±1.51 | 57.4±0.44 | 0.63±0.25 | 53.7±2.11 | 45.6±1.18 | 0.70±0.11 | 65.7±1.35 | 37.3±0.26 | 0.63±0.10 | 73.0±0.61 |
| HCC | 68.4±1.73 | 0.82±0.21 | 41.3±2.52 | 56.7±0.79 | 1.04±0.12 | 50.5±0.76 | 44.9±1.60 | 0.77±0.11 | 66.2±1.60 | 35.8±0.65 | 0.84±0.15 | 73.5±1.14 |

Table 2: Performance comparison of different models on the Drebin dataset for offline learning. The results represent the mean and standard deviation over five runs with different random seeds. The best-performing model is underlined.

| Model | Merged Training | | | | | | Holdout Training | | | | | |
|---|---|---|---|---|---|---|---|---|---|---|---|---|
| | Duplicated | | | Deduplicated | | | Duplicated | | | Deduplicated | | |
| | F1 | FPR | FNR | F1 | FPR | FNR | F1 | FPR | FNR | F1 | FPR | FNR |
| RF | 64.6±1.02 | 0.23±0.00 | 49.6±1.08 | 61.3±0.92 | 0.50±0.03 | 51.7±0.79 | 45.3±5.22 | 0.12±0.02 | 68.5±4.50 | 46.8±3.89 | 0.34±0.08 | 67.0±3.76 |
| SVM | 76.0±0.00 | 1.20±0.00 | 30.7±0.00 | 69.7±0.00 | 1.41±0.00 | 34.9±0.00 | 74.5±0.00 | 0.86±0.00 | 34.7±0.00 | 66.3±0.00 | 1.42±0.00 | 39.5±0.00 |
| XGBoost | 74.6±0.47 | 1.21±0.09 | 32.5±0.49 | 69.0±0.45 | 1.61±0.03 | 34.3±0.57 | 69.5±1.40 | 1.03±0.06 | 40.2±1.65 | 65.2±0.62 | 1.33±0.05 | 41.6±0.88 |
| MLP | 68.6±2.28 | 0.85±0.26 | 41.9±3.77 | 63.0±2.73 | 0.96±0.37 | 46.32±4.9 | 55.6±4.26 | 0.90±0.20 | 56.2±4.85 | 57.0±3.52 | 1.67±0.64 | 49.9±6.78 |
| SCC | 66.1±2.79 | 1.06±0.26 | 43.8±3.94 | 66.2±2.09 | 2.11±0.57 | 35.2±1.98 | 70.36±2.9 | 1.79±0.16 | 35.3±3.33 | 57.0±3.52 | 1.67±0.64 | 49.9±6.78 |
| HCC | 73.3±2.37 | 1.35±0.15 | 33.3±3.90 | 66.0±1.75 | 0.92±0.17 | 42.8±2.66 | 64.2±5.17 | 1.60±0.20 | 43.8±6.61 | 62.6±2.74 | 1.59±0.26 | 43.8±4.13 |

Table 3: Performance comparison of different models on the APIGraph Dataset for offline learning

2. **Benefits of merged training:** Merging validation data with training data before final evaluation consistently enhances performance. In the Drebin deduplicated dataset, the average F1-score improvement ranges from 8.3 (RF) to 20.9 (HCC), demonstrating the benefits of additional training data and its alignment with practical applications.

3. **Duplicates influence model preference:** Duplicates can skew model comparisons. For example, in the Drebin dataset merge-training setting, SCC outperforms XGBoost by 11 in average F1-score. However, after deduplication, XGBoost outperforms SCC by 2.1 in average F1-score.

4. **XGBoost as a strong baseline on deduplicated datasets:** With proper tuning, XGBoost achieves the highest average F1-score on the deduplicated Drebin dataset and remains competitive on APIGraph, highlighting its effectiveness as a baseline model. In contrast, while SVM performs well on APIGraph, its performance drops significantly on the Drebin dataset after deduplication.

## 6.2 Duplicates and Variance in Performance

Figures 3 and 4 show the month-by-month performance of four models (SVM, XGBoost, MLP, HCC) on the APIGraph and Drebin datasets under the merged training setting for both duplicated and deduplicated datasets. Key conclusions from the results are:

1. **Impact of duplicates on performance trends:** On both datasets, models generally exhibit a downward performance trend over time due to concept drift. However, this trend is less evident in the original datasets with duplicates, where sudden fluctuations occur. For example, all models experienced a performance drop in November 2015 on the APIGraph dataset, explained by duplicate inputs, as shown in Figure 5 (left). Of the 488 malware samples that month, 190 have identical input features. If a model fails to detect one correctly, it fails for all, resulting in a significant performance drop.

2. **High variance due to duplicates:** As seen in Figure 4(b), duplicates can cause high variance in model performance. Figure 4(c) shows that this effect is primarily due to variance introduced by false negative rates (FNRs). Figure 5(right) illustrates the frequency of unique, and top-10 duplicated

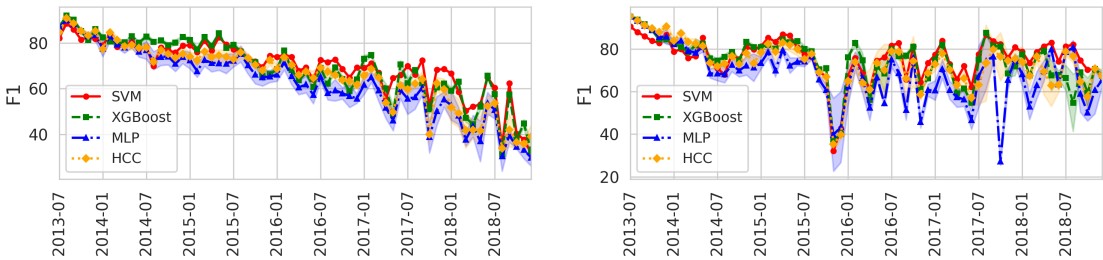

Figure 3: F1-score over the test months on the APIGraph for (left) deduplicated (right) duplicated datasets

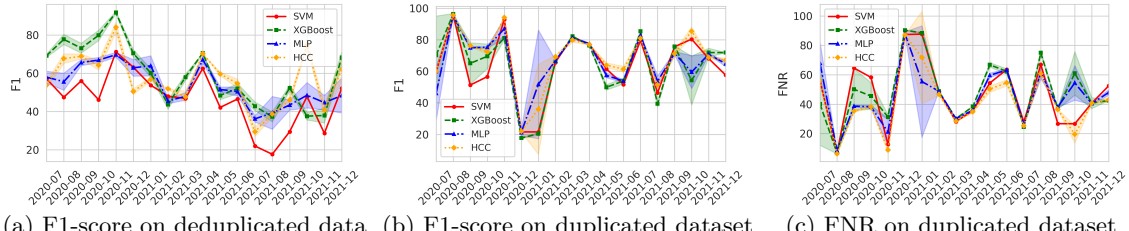

(a) F1-score on deduplicated data  (b) F1-score on duplicated dataset  (c) FNR on duplicated dataset

Figure 4: Performance in test months on the Drebin datasets for Merged Training setting

malware samples for January 2021 on the Drebin dataset. Specifically, 348 out of 385 samples are duplicates of the same input feature, resulting in significant variance for models relying on random initialization, such as neural networks. Two of five MLPs trained with different seeds achieved high F1 scores (93.7 and 94.0), while the others scored much lower (21.4, 24.6, 24.8), indicating high fluctuations due to duplicates. A theoretical classifier correctly identifying all malware samples this month has an FNR of 0. If it fails to detect this single duplicated sample, the FNR rises to 90.4%. Conversely, a classifier detecting no samples has an FNR of 100%, but identifying only the duplicated sample reduces the FNR to 9.6%. This highlights the need to deduplicate datasets for models relying on random initialization to prevent performance disparities caused by different initializations.

3. **Importance of reporting performance with multiple seeds:** Although deduplication reduces extreme variance in model performance for certain months, neural networks can still show performance variation due to random initialization, as seen in Figure 4(a). For instance, in four test months (2021-01, 2021-08, 2021-10, 2021-12), the MLP had a standard deviation of average F1-scores greater than 5, even on the deduplicated Drebin dataset. Therefore, accounting for this randomness when reporting neural network performance is essential.

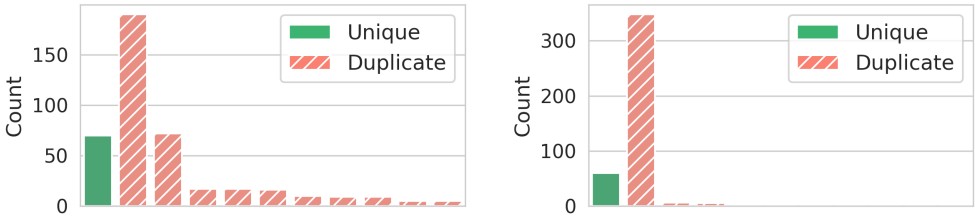

Figure 5: Frequency of Unique and top-10 Duplicate malware samples for the month 2015-11 on the API-Graph dataset (left) and the month 2021-01 on the Drebin dataset (right)

| Model | Budget=50 | | | | | | Budget=100 | | | | | |
|---|---|---|---|---|---|---|---|---|---|---|---|---|
| | Merged Training | | | Holdout Training | | | Merged Training | | | Holdout Training | | |
| | F1 | FPR | FNR | F1 | FPR | FNR | F1 | FPR | FNR | F1 | FPR | FNR |
| RF | 61.9±0.73 | 0.13±0.02 | 53.0±0.88 | 62.7±1.11 | 0.11±0.00 | 52.4±1.11 | 62.8±1.24 | 0.13±0.01 | 52.1±1.30 | 63.6±0.47 | 0.12±0.03 | 51.3±0.40 |
| SVM | 64.4±0.00 | 0.22±0.00 | 48.7±0.00 | 59.6±0.00 | 0.23±0.00 | 54.1±0.00 | 68.1±0.00 | 0.24±0.00 | 44.6±0.00 | 63.9±0.00 | 0.26±0.00 | 49.1±0.00 |
| XGBoost | 82.2±0.60 | 0.15±0.00 | 27.3±0.76 | 81.1±0.15 | 0.14±0.01 | 29.2±0.21 | 82.5±0.29 | 0.15±0.01 | 26.9±0.32 | 82.5±0.45 | 0.14±0.01 | 27.0±0.57 |
| MLP | 67.9±0.65 | 0.49±0.03 | 43.3±0.70 | 61.8±0.54 | 0.35±0.02 | 51.0±0.49 | 70.2±0.42 | 0.38±0.03 | 40.6±0.46 | 65.7±3.37 | 0.36±0.06 | 46.2±4.17 |
| SCC | 71.0±0.64 | 0.59±0.04 | 37.4±0.29 | 68.2±1.83 | 0.50±0.07 | 41.6±2.21 | 73.7±1.20 | 0.57±0.10 | 33.7±1.35 | 72.1±0.88 | 0.48±0.07 | 37.0±1.44 |
| HCC | 73.5±0.27 | 0.66±0.04 | 32.8±0.85 | 68.5±3.38 | 0.57±0.05 | 40.3±4.37 | 75.6±0.96 | 0.68±0.06 | 29.4±1.07 | 72.6±2.10 | 0.57±0.02 | 35.2±2.64 |

Table 4: Performance of different models on the Drebin Dataset for continuous active learning with 50 & 100 annotation budget per month

| Model | Budget=50 | | | | | | Budget=100 | | | | | |
|---|---|---|---|---|---|---|---|---|---|---|---|---|
| | Merged Training | | | Holdout Training | | | Merged Training | | | Holdout Training | | |
| | F1 | FPR | FNR | F1 | FPR | FNR | F1 | FPR | FNR | F1 | FPR | FNR |
| RF | 87.1±0.15 | 0.27±0.00 | 19.4±0.28 | 87.4±0.20 | 0.30±0.01 | 18.6±0.34 | 89.0±0.11 | 0.26±0.01 | 16.4±0.13 | 89.4±0.11 | 0.28±0.01 | 15.6±0.11 |
| SVM | 83.7±0.00 | 0.60±0.00 | 21.4±0.00 | 84.4±0.00 | 0.67±0.00 | 19.2±0.00 | 84.8±0.00 | 0.63±0.00 | 19.0±0.00 | 85.2±0.00 | 0.75±0.00 | 16.9±0.00 |
| XGBoost | 88.7±0.21 | 0.55±0.01 | 13.8±0.31 | 88.9±0.11 | 0.53±0.03 | 13.7±0.11 | 90.4±0.13 | 0.48±0.01 | 11.6±0.21 | 90.6±0.18 | 0.45±0.01 | 11.5±0.16 |
| MLP | 85.7±0.67 | 0.47±0.03 | 19.6±1.12 | 85.7±0.40 | 0.47±0.05 | 19.5±1.10 | 87.4±0.29 | 0.47±0.02 | 16.6±0.49 | 87.2±0.09 | 0.49±0.04 | 16.6±0.60 |
| SCC | 85.6±0.37 | 0.85±0.07 | 15.1±0.27 | 86.0±0.71 | 0.81±0.08 | 14.8±0.31 | 86.1±0.38 | 0.95±0.07 | 12.8±0.61 | 85.7±2.80 | 2.03±2.51 | 12.9±0.48 |
| HCC | 87.1±0.36 | 0.64±0.05 | 15.6±0.95 | 86.7±0.15 | 0.56±0.03 | 17.2±0.36 | 87.5±0.78 | 0.56±0.03 | 15.5±1.11 | 87.4±0.27 | 0.59±0.02 | 15.6±0.14 |

Table 5: Performance of different models on the APIGraph Dataset for continuous active learning with 50 & 100 annotation budget per month

### 6.3 Effective Model Selection for Robust Baselines

Our experiments with offline learning show that the XGBoost model achieves superior performance across various experimental settings, contrasting with previous results from Chen et al. (2023a). The key reason is the expanded hyperparameter search space. We included more hyperparameters and broadened the search range, as detailed in Appendix C. A critical change was the number of boosting rounds considered. Chen et al. (2023a) tested rounds ranging from 10 to 100. However, XGBoost often benefits from more boosting rounds, particularly on the APIGraph dataset, where 400 rounds yielded the best results for duplicated and deduplicated datasets. This underscores the importance of proper hyperparameter tuning to establish stronger baseline models (Arp et al., 2022).

## 7 Continuous Active Learning Results & Analysis

In the continuous active learning setting, duplicates pose additional challenges beyond those in offline learning. Since active learning relies on human experts to select samples for annotation, increasing the diversity of selected samples is crucial (Settles, 2009). Duplicates can lead to redundant selections, wasting the annotation budget. A simple heuristic can identify previously annotated samples and replace them with their existing annotations. Thus, using duplicated datasets in an active learning setting is impractical, and our analysis primarily focuses on deduplicated datasets.

We present the results of the six models in Tables 4 and 5 for the Drebin and APIGraph datasets, discussing merged and holdout training settings. Key findings are summarized below:

1. **XGBoost as a strong baseline in continuous active learning:** XGBoost achieves the highest average F1 score across all experimental settings in continuous active learning. This is especially evident in the challenging Drebin dataset, where it surpasses the second-best model by 8.7% in the merged training setting. This highlights the importance of selecting a strong baseline model and underscores XGBoost's effectiveness in Android malware detection.

2. **Benefits of Contrastive Learning for Neural Networks:** The neural networks trained with contrastive learning, SCC, and HCC outperform baseline neural networks in all settings, indicating

the effectiveness of contrastive learning in enhancing robustness to concept drift. HCC further outperforms SCC, demonstrating the advantages of hierarchical contrastive learning and the pseudo-loss sample selector introduced in Chen et al. (2023a). However, more improvements are needed to match XGBoost's performance.

3. **Random Forest as a Competitive Baseline on APIGraph:** Even simpler tree-based methods like Random Forest achieve competitive performance on the APIGraph dataset, often surpassing more complex neural networks. With consistently low false positives, RF is a strong candidate model for this dataset, illustrating that simpler models can compete effectively with complex ones in a compact feature space.

4. **Performance Gains from Merged Training:** Merging the validation set with the training set can enhance performance on test data, particularly for neural networks. However, the performance difference is less pronounced than offline learning since models are retrained during validation months with a subset of samples.

## 8 Limitations & Discussions

In our study, we define duplicates as instances with identical feature inputs to the model. However, this does not necessarily correspond to true duplicates in the input space due to factors such as obfuscation, packing, feature space drift, and other complications (O'Kane et al., 2011; Yang et al., 2015; Chen et al., 2023b). Deduplicating malware is, therefore, a challenging task and is not the primary focus of our work. Instead, our objective is to evaluate the reproducibility of reported results when using identical feature sets and machine learning models. It is important to exercise caution when interpreting these results. For example, one feature set may treat near-duplicate malware as identical, while another feature set may introduce variability between them. However, this variability is a property of the feature extraction process, distinct from the machine learning model. Consequently, we argue that deduplication is necessary when comparing different machine learning models on the same datasets, or at the very least, the variance in performance should be explicitly addressed.

Our study examined two datasets with static feature sets, although other feature sets and datasets have been proposed in the Android malware detection literature (Liu et al., 2020). While dynamic analysis may be required for more robust malware detection, our findings regarding model selection, robust baselines, and evaluation strategies apply to other feature sets and can inform future research. Although duplicates and deduplication processes may vary across feature sets, the broader implications of our analysis remain relevant.

Datasets with labels aggregated from sources such as VirusTotal may suffer from issues related to ground truth annotations, as highlighted in prior studies (Aghakhani et al., 2020). However, addressing these issues is beyond the scope of our study. Our focus is on offline and continuous active learning for malware detection. In the continuous learning setting, we retrain the model monthly with new samples, as done in prior work. However, this approach may be inefficient if the model's performance has not degraded. In such cases, a drift detection mechanism may be necessary to determine when retraining is required (Jordaney et al., 2017).

## 9 Conclusion

In this paper, we have critically examined the challenges of reproducibility and replicability in Android malware detection, focusing on key pitfalls in dataset curation, model selection, and evaluation strategies. Our extensive evaluation across multiple machine learning models in offline and continuous active learning settings reveals that issues such as data duplication, inadequate hyperparameter tuning, and biased model selection can lead to significant performance disparities. These findings highlight the need for more rigorous approaches to ensure the reliability of reported results. Our results indicate that, with proper tuning, simpler models can often match or surpass the performance of more complex methods, challenging the trend of relying solely on sophisticated architectures. Our open-source code offers a standardized, extensible framework for benchmarking new models, facilitating fairer and more reliable comparisons across studies.

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

# A  Machine Learning Models

We utilize six different models commonly used in prior works on Android malware detection:

1. **Random Forest (RF):** Random Forest (Breiman, 2001) is an ensemble learning method for classification that constructs multiple decision trees during training and outputs the mode of their classes. It has been used in Android malware detection (Sanz et al., 2013; Mariconti et al., 2017).

2. **Support Vector Machine (SVM):** SVM (Cortes, 1995) is a classification algorithm that finds the hyperplane that best separates classes in the feature space. It has been widely used for Android malware detection, particularly with Drebin features (Arp et al., 2014; Jordaney et al., 2017).

3. **eXtreme Gradient Boosting (XGBoost):** XGBoost (Chen & Guestrin, 2016) is a scalable and efficient gradient boosting method that excels in performance on various tabular datasets (Grinsztajn et al., 2022), although it has not shown consistent success in Android malware detection (Chen et al., 2023a).

4. **Multilayer Perceptron (MLP):** An MLP is a type of neural network composed of multiple layers of neurons, with each layer fully connected to the next (Rumelhart et al., 1986). MLPs are used for a variety of tasks, including classification and regression, and have been employed in several prior works for Android malware detection (Pendlebury et al., 2019; Yuan et al., 2016; Kim et al., 2018).

5. **Supervised Contrastive Classifier (SCC):** Contrastive learning effectively detects and adapts to concept drift in Android malware detection (Yang et al., 2021; Chen et al., 2023a). This approach extracts meaningful representations by distinguishing between similar and dissimilar instance pairs. It assumes that similar instances should be closer in the embedding space while dissimilar ones should be further apart (Wu et al., 2018; Khosla et al., 2020).

   We implement a method based on triplet learning, focusing on the main classes—benign and malware—with a triplet loss. In our approach, a sample is considered positive if it belongs to the same class as the anchor and negative if it belongs to a different class. Unlike previous works that treat each malware family as a distinct class, leading to $m + 1$ classes where $m$ is the number of malware families, our method only considers the binary case. This is more annotation-friendly, as it does not require separate malware family annotation.

   The learning process combines a supervised binary cross-entropy loss and a triplet margin loss. The binary cross-entropy loss is defined as:

   $$\mathcal{L}_{\text{bce}} = -\frac{1}{N} \sum_{i=1}^{N} [y_i \log(\hat{y}_i) + (1 - y_i) \log(1 - \hat{y}_i)]$$

   where $y_i$ is the true label and $\hat{y}_i$ is the predicted probability.

   The triplet margin loss is given by:

   $$\mathcal{L}_{\text{triplet}} = \frac{1}{N} \sum_{i=1}^{N} \max\left(0, \|\text{enc}(x_i^a) - \text{enc}(x_i^p)\|_2^2 - \|\text{enc}(x_i^a) - \text{enc}(x_i^n)\|_2^2 + \text{m}\right)$$

   where $x_i^a, x_i^p, x_i^n$ are the anchor, positive, and negative samples respectively, $\text{enc}(\cdot)$ is the encoder function, and $m$ is the margin parameter.

   The combined loss function is:

   $$\mathcal{L} = \lambda \cdot \mathcal{L}_{\text{bce}} + \mathcal{L}_{\text{triplet}}$$

   where $\lambda$ is a weight parameter balancing the two loss components.

6. **Hierarchical Contrastive Classifier (HCC):** Chen et al. (2023a) introduced an enhanced supervised contrastive classifier featuring a hierarchical contrastive loss function. The core idea is to ensure that samples from the same malware family are more similar than those from different families in the feature embedding space. Using an encoder network enc, the embeddings of two benign or two malicious samples from different families should satisfy $\|enc(x_1) - enc(x_2)\|_2 \leq m$. For samples from the same malware family, the embeddings should be even more tightly constrained, while embeddings of one malicious and one benign sample should be highly dissimilar, $\|enc(x_1) - enc(x_2)\|_2 \geq 2m$.

Let $d_{ij}$ denote the Euclidean distance between two samples $i$ and $j$ in the embedding space, defined as $d_{ij} = \|enc(x_i) - enc(x_j)\|_2$. The hierarchical contrastive loss is:

$$\mathcal{L}_{hc}(i) = \frac{1}{|P(i, y_i, y_i')|} \sum_{j \in P(i, y_i, y_i')} \max(0, d_{ij} - m)$$
$$+ \frac{1}{|P_z(i, y_i, y_i')|} \sum_{j \in P_z(i, y_i, y_i')} d_{ij}$$
$$+ \frac{1}{|N(i, y_i)|} \sum_{j \in N(i, y_i)} \max(0, 2m - d_{ij})$$

The first term ensures positive pairs, such as (benign, benign) or (malicious, malicious), are close but not overly constrained, penalizing distances only if they exceed $m$. The second term enforces similarity within the same malware family. The last term aims to separate benign and malicious samples by at least $2m$.

Like the SCC, this loss is combined with binary cross-entropy loss to train the model. Note that this is the only model that requires access to malware family labels to train the model.

## B    Implementation Details

### B.1    Environment

We conducted all experiments using Python 3.11.6 on Red Hat Enterprise Linux (Version 9.4). Neural network training was performed using a single NVIDIA A100 GPU. Table 6 lists the Python pip packages used in the implementation:

| Pip Package | Version | Utility |
|---|---|---|
| scikit-learn | 1.5.0 | Used to train RF, SVM |
| torch | 2.1.0 | Used to train neural networks (MLP, SCC, HCC) |
| xgboost | 2.1.0 | Used to train XGBoost |
| numpy | 1.26.4 | Used to process datasets |
| pytorch-metric-learning | 2.6.0 | Used for creating data sampler for HCC |
| yacs | 0.1.8 | Configuration management for experiments |

Table 6: Python pip packages used in the implementation

### B.2    Code Structure

The code is organized into multiple key modules to facilitate easy extension:

1. **Dataset:** We preprocess data into a single `.npz` file containing features, labels, timestamps, and duplication indicators. The `intra_split_dupes` field marks duplicate within the current timestamp,

referring to earlier indices or set to `-1` if unique. The `cross_split_dupes` field tracks duplicates across splits, indicating the relevant split and index or `none` if unique. This format supports active and offline learning deduplication. The `MalwareDataset` class handles dataset loading and includes the HalfSampler method used in HCC (Chen et al., 2023a), as well as a class for working with Triplet Datasets.

2. **Models:** This module implements six models, all inheriting from a `BaseModel` class, which takes `params` for hyperparameters and `cfg` for experiment-specific settings (e.g., output classes, GPU usage). Each model includes methods for fitting (`fit`), prediction (`predict`, `predict_proba`), and active learning sample selection (`sample_active_learning`).

3. **Tasks:** We separate offline and active learning into distinct tasks within the `tasks` module, each with specific training and evaluation procedures. Tasks use a `yacs` configuration (Girshick, 2020), provided as a YAML file, specifying the task type, model, and relevant settings. This separation allows easy extension for other malware analysis tasks, such as family classification or concept drift detection, with minimal code changes.

## C Hyperparameter Search

### C.1 Hyperparameter Search-Space

The detailed hyperparameter search space is shown in Table 7. Most hyperparameter names follow standard terminologies in the scikit-learn, XGBoost, or PyTorch APIs. Below, we describe those with different or new terminology:

- **XGBoost/MLP/SCC - class-weight:** This parameter, "scale_pos_weight" in XGBoost, defaults to 1 if set to False. When True, it adjusts for class imbalance by weighting the positive class (malware) according to the benign-to-malware ratio in the training data. For MLP and SCC, samples are either balanced in each batch with equal benign and malware samples (balance=True) or drawn randomly (balance=False).

- **MLP/SCC/HCC - cont_learning_epochs:** In continuous active learning with neural networks, initial and retraining epochs differ, similar to Chen et al. (2023a). For HCC, it specifies the number of epochs; for MLP and SCC, it represents the fraction of the original training epochs.

- **HCC - cont_learning_lr:** For HCC, the initial learning rate is varies the learning rate for each update, while MLP and SCC continue with the current rate using the Adam optimizer.

- **SCC/HCC - xent_lambda:** This controls the binary cross-entropy loss weight in contrastive learning. We use a value of 100, as suggested in Chen et al. (2023a).

- **SCC/HCC - margin:** This sets the margin for computing contrastive loss. We use a value of 10, following Chen et al. (2023a).

We adopt strategies from prior works on Android malware detection (Chen et al., 2023a), machine learning on tabular data (McElfresh et al., 2024), and deep neural architectures (Bengio, 2012) to design the hyperparameter search space.

While we mostly follow the hyperparameters for HCC from Chen et al. (2023a), there are a few minor differences. These choices ensure consistency with other neural network architectures and ease of implementation. We explore two architectures for the encoder and MLP layers, each instead of the single one used in their work. We include dropout as a hyperparameter instead of setting it to a fixed value of 0.2 in their study. Only the Adam optimizer is used during the continuous learning retraining phase. We employ a step learning rate scheduler and omit the cosine annealing scheduler. Regardless, we verify that our search space includes the best hyperparameter set reported in Chen et al. (2023a); for instance, cosine annealing did not outperform step-based learning rates in any dataset in their results.

| Model | Hyperparameter | Candidate Values |
|---|---|---|
| Random Forest | n_estimators | $2^x$, where $x \in \mathrm{U}[5, 10]$ |
| | max_depth | $2^y$, where $y \in \mathrm{U}[5, 10]$ |
| | criterion | {gini, entropy, log_loss} |
| | class_weight | {None, "balanced"} |
| SVM | C | $10^z$, where $z \in \mathrm{U}[-4, 3]$ |
| | class_weight | {None, "balanced"} |
| XGBoost | max_depth | $2^w$, where $w \in \mathrm{U}[3, 7]$ |
| | alpha | $10^a$, where $a \in \mathrm{U}[-8, 0]$ |
| | lambda | $10^b$, where $b \in \mathrm{U}[-8, 0]$ |
| | eta | $3.0 \times 10^c$, where $c \in \mathrm{U}[-2, -1]$ |
| | balance | {True, False} |
| | num_boost_round | {100, 150, 200, 300, 400} |
| | subsample | $x$, where $x \in \mathrm{U}[0.8, 1.0]$ |
| | colsample_bytree | $x$, where $x \in \mathrm{U}[0.8, 1.0]$ |
| MLP | mlp_layers | {[100, 100], [512, 256, 128], [512, 384, 256, 128], [512, 384, 256, 128, 64]} |
| | learning_rate | $10^d$, where $d \in \mathrm{U}[-5, -3]$ |
| | dropout | $x$, where $x \in \mathrm{U}[0.0, 0.5]$ |
| | batch_size | $2^e$, where $e \in \{5, 6, 7, 8, 9, 10\}$ |
| | epochs | {25, 30, 35, 40, 50, 60, 80, 100, 150} |
| | optimizer | {Adam} |
| | balance | {True, False} |
| | cont_learning_epochs* | {0.1, 0.2, 0.3, 0.4, 0.5} |
| SCC | encoder_layers | {[512, 256, 128], [512, 384, 256, 128]} |
| | mlp_layers | {[100], [100, 100]} |
| | learning_rate | $10^f$, where $f \in \mathrm{U}[-5, -3]$ |
| | dropout | $y$, where $y \in \mathrm{U}[0.0, 0.25]$ |
| | batch_size | $2^g$, where $g \in \{9, 10, 11\}$ |
| | epochs | {25, 30, 35, 40, 50, 60, 80, 100} |
| | xent_lambda | {100} |
| | margin | {10} |
| | optimizer | {Adam} |
| | balance | {True, False} |
| | cont_learning_epochs* | {0.1, 0.2, 0.3, 0.4, 0.5} |
| HCC | encoder_layers | {[512, 256, 128], [512, 384, 256, 128]} |
| | mlp_layers | {[100], [100, 100]} |
| | learning_rate | {0.001, 0.003, 0.005, 0.007} |
| | dropout | $z$, where $z \in \mathrm{U}[0.0, 0.25]$ |
| | batch_size | $2^{10}$ |
| | epochs | {100, 150, 200, 250} |
| | xent_lambda | {100} |
| | margin | {10} |
| | optimizer | {Adam, SGD} |
| | scheduler_step | {10} |
| | scheduler_gamma | {0.5, 0.95} |
| | cont_learning_lr* | {0.01, 0.05} |
| | cont_learning_epochs* | {50, 100} |

Table 7: Hyperparameter Search Spaces for Different Models. U indicates sampling from a uniform random distribution over values in the range. * indicates hyperparameters that are only used during continuous learning retraining phase.

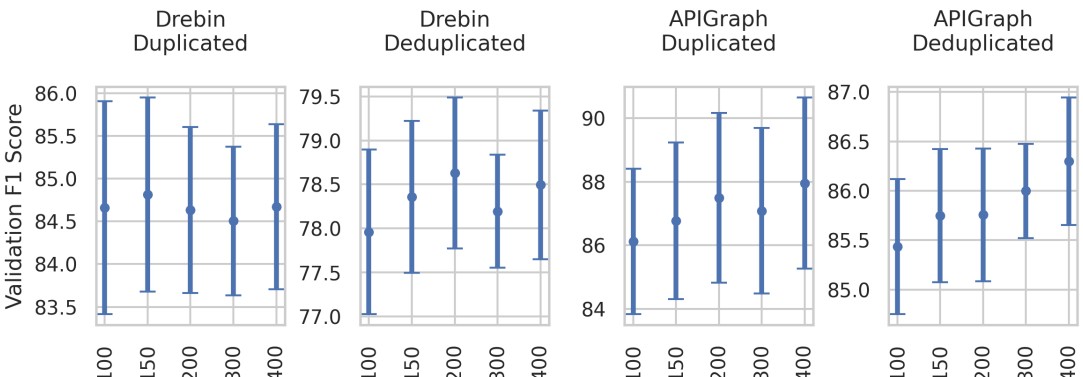

Figure 6: Effect of number of boosting rounds during learning for XGBoost model on different datasets

### C.2 Impact of Boosting Rounds on XGBoost Performance

Figure 6 illustrates how the number of boosting rounds affects XGBoost's performance on the validation dataset across different datasets in an offline learning setup.

### C.3 Number of Trials and Performance Convergence

We analyze model performance on the validation set as the number of hyperparameter search trials increases across different datasets. Figure 7 presents results for the offline learning setting, while Figure 8 shows results for continuous active learning. Most models converge within 100 trials in offline learning, whereas convergence occurs even earlier in continuous learning, indicating that initial hyperparameters have less impact. Although validation performance generally reflects model effectiveness, this is not always the case. For instance, the RF model performs significantly better on the validation set than on the test set for the Drebin dataset in both settings.

## D  Duplicates & Model Selection

The presence of duplicates can influence hyperparameter tuning and model selection. Figure 9 demonstrates this for XGBoost by adjusting the "scale_pos_weight" parameter to address class imbalance. In the imbalanced setting, the malware class weight is set to 1, while in the balanced setting, it matches the benign-to-malware ratio in the training set. Balancing improves validation performance for duplicated datasets but may reduce performance on the deduplicated APIGraph dataset, suggesting that duplicates can alter hyperparameter behavior for the same model.

Figure 10 illustrates how duplicates affect feature weighting in XGBoost. It compares feature importance between models trained on the deduplicated and original duplicated Drebin datasets. The top 50 features from the deduplicated model were extracted and their importance evaluated in the duplicated model. The results show a clear disparity in feature importance distribution between the two models.

## E  Variance in Results for Continuous Learning

Figure 11 shows the monthly F1 scores on the Drebin dataset in continuous active learning settings, comparing three neural networks on both duplicated and deduplicated datasets. Separate hyperparameter tuning was performed for MLP and SCC, while for HCC, we used the artifacts provided in Chen et al. (2023a) for the duplicated setting to maintain consistency with their study. Notably, we observe extreme variance in model performance in certain months (e.g., December 2020) in the duplicated datasets.

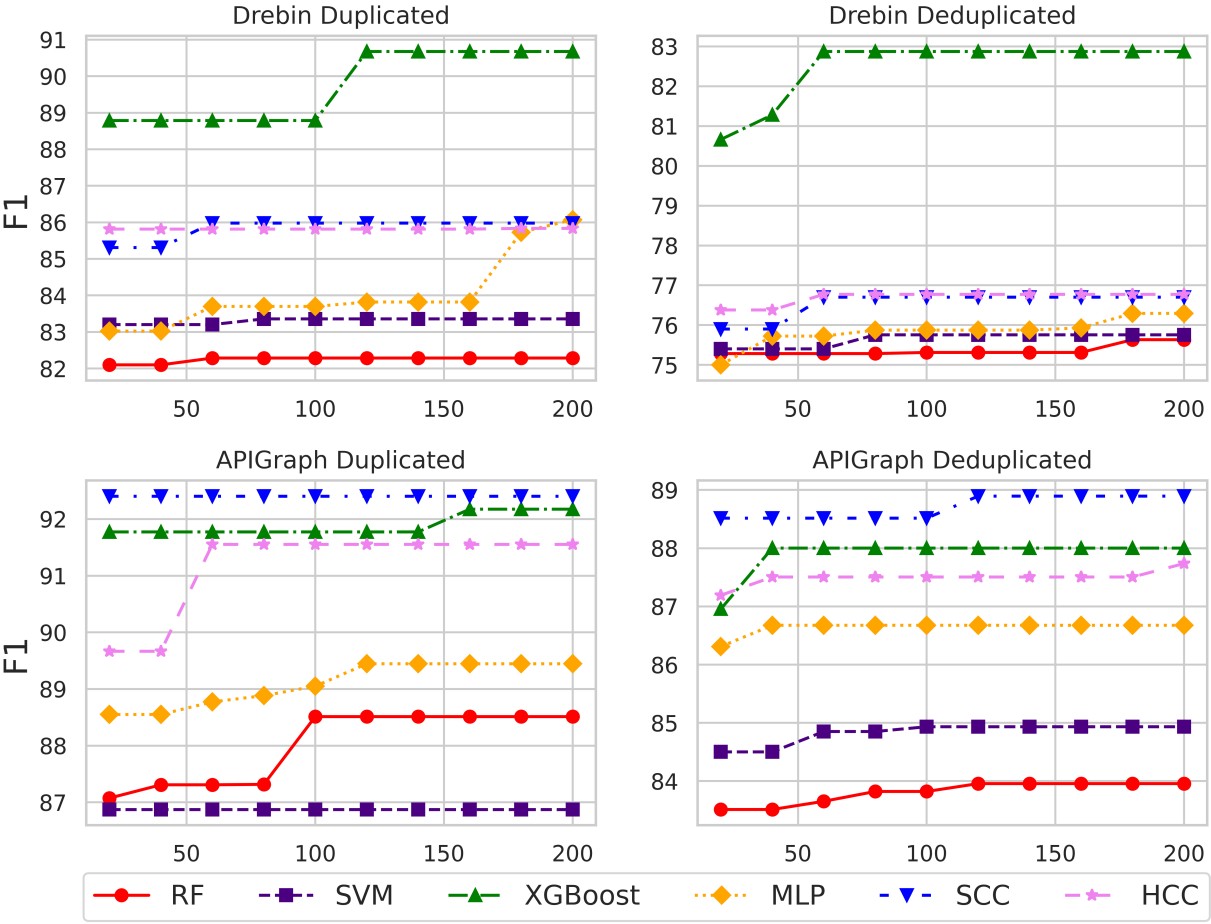

Figure 7: Performance on validation set as number of hyperparameter trials is increased for offline learning

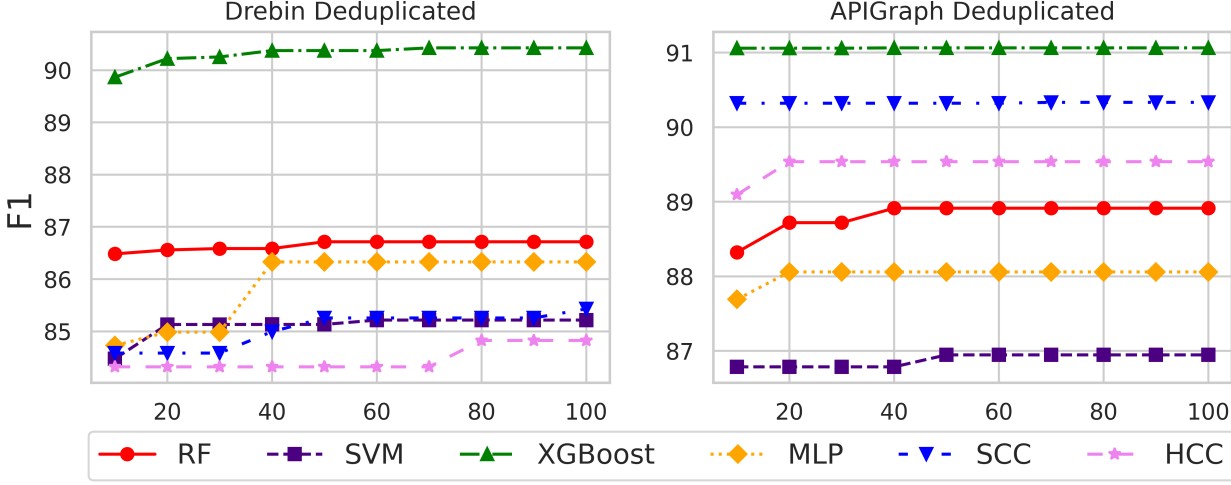

Figure 8: Performance on validation set as number of hyperparameter trials is increased for continuous active learning

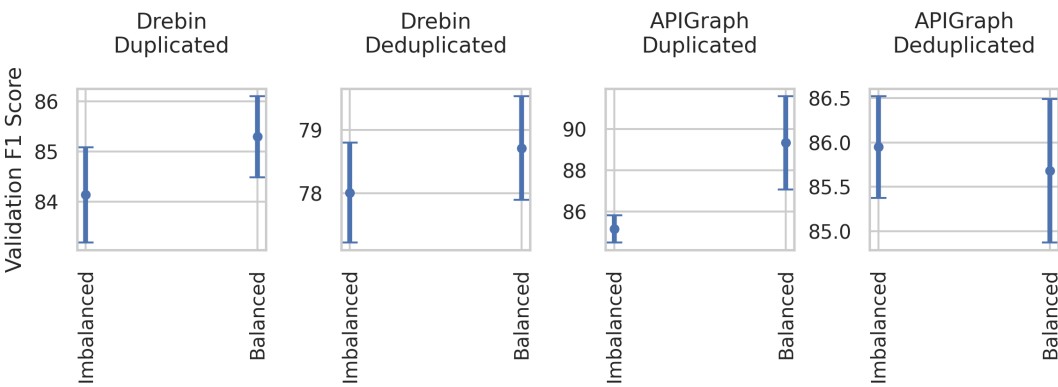

Figure 9: Effect of balancing the weight of the classes during learning for XGBoost model on different datasets

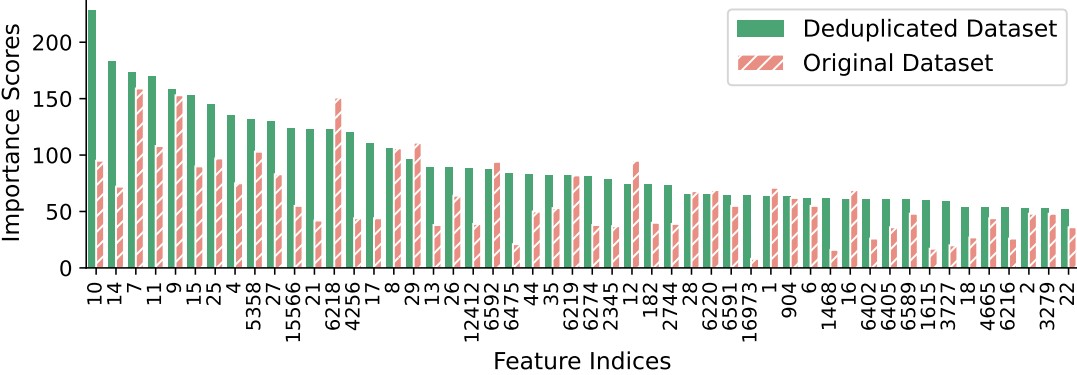

Figure 10: Comparison of feature weights for the top 50 features in XGBoost models trained on the deduplicated Drebin dataset and their corresponding weights in the model trained on the original duplicated dataset.

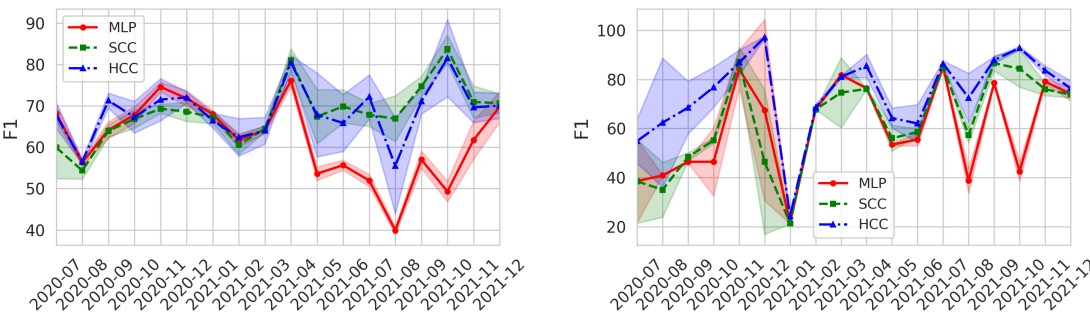

Figure 11: F1-score over the test months for active learning on the (left) deduplicated (right) duplicated Drebin datasets

