# OpenReview forum: "Revisiting Static Feature-Based Android Malware Detection using Machine Learning"
_TMLR — Rejected by TMLR_

### Review · Reviewer_hGqU · 2025-02-12

**Summary Of Contributions:**

This work presents an experimental study on model selection for malware detection tasks, focusing on Android malware samples and a variety of machine learning classifiers ranging from tree-based methods to more complex neural network-based ones.
Specifically, it focuses on reproducibility problems that hinder a considerable portion of machine-learning-based approaches for Android malware detection. It primarily focuses on reproducing the results of one paper, which is considered state-of-the-art in Android malware detection. For completeness, the paper also focuses on five other approaches based on more simplistic classifiers, such as Random Forest and Support vector machine-based ones, totaling in a number of 6 methods that are benchmarked and evaluated. These techniques are evaluated in both the offline and online learning settings. The main contribution of this work is the analysis of prior work with the scope of proving insights about the importance of dataset quality/hyperparameter tunning and their impact on the classifier's performance.

**Audience:**

Yes

**Broader Impact Concerns:**

No concerns

**Claims And Evidence:**

Yes

**Requested Changes:**

I believe that an expansion of the work in evaluating other approaches in different settings and different datasets would improve the work from the experimental point of view. I would also suggest the authors provide, where possible, better insights regarding the obtained results of the experiments that they have carried out. Refer to the Weaknesses comments above for more.

**Strengths And Weaknesses:**

# Strengths of the work:

The work is generally well-written and easy to follow, given that the sections and their division have a logical and consistent flow of information. The code repository is decently organized.

# Weaknesses of the work
As mentioned above, the paper is well-written, but what feels lacking is more commenting on the insights obtained from the experimental evaluation, not just stating the apparent result (i.e., saying this is better than the other) but a more profound discussion (and possible exploration) on why it happens would be much appreciated.

- The number of datasets:
I would expect a larger number of datasets to be included in the study in order to gain a more thorough insight into the dataset implications for the resulting classifiers.

- The duplicates selection approach:
The duplicate selection and removal approach raises concerns about its efficiency and applicability to the Android malware detection use cases discussed in the paper. For example, if the feature extraction process removes some important distinctions, we would end up with non-similar raw samples being mapped to the same features. Moreover, depending on the nature of the data, assuming numerical data, small variations might lead to the same features but not necessarily this would mean the same functionality. Other concerns are from the semantical point of view, where two different samples might have the same representation. In the case of high-dimensional data, some of these intrinsic features are learned automatically through the learning classifier. They would be better left in the training set for the model to discover the distinctions, not be removed beforehand automatically.

- Insights about the seed selection and reporting:
The paper highlights that research should provide the results of an ML-based detection approach using different seeds, highlighting that the seed is an important hyperparameter in model training. It is commonly known that the seed is an important hyperparameter, not just for the reproducibility of the works but also for the initialization of the weights and biases in the case of deep learning-based classifiers might lead to different outcomes (better or worse), keeping fixed the other parameters such as training epochs, learning rates training set size, etc. I agree that research works might benefit by publishing the results with different seeds, but this is not crucial. The overall approach is essential, as is dataset preprocessing, architecture selection, etc. The hyperparameters are an important factor, but parameters such as the initialization seed are purely based on luck, more than proper and consolidated insights that would correspond to better/worse performance of the model. What could be done is having the seed as just another parameter to search following a Grid search, but that's just about it.

 - The key takeaway:
The takeaway this paper provides is the fact that simpler baseline models perform better than complex ones when appropriately trained. This is commonly accepted and observed in almost all the academic approaches that rely on ML techniques for different types of malware detection, such as Ransomware, keyloggers, etc. Generally, approaches that rely on simplistic architectures, such as random forests, are shown to perform better and also be more efficient to run in real-time detection scenarios. In essence, this is due to the underlying (simplistic) complexity of the datasets upon which the models are trained.

---

> ### Author Response · Authors · 2025-02-26
> **Response to Reviewer hGqU**
>
> We sincerely appreciate the reviewer's insightful suggestions and comments.
>
> **Response to Weaknesses**
>
> - **The number of datasets:** One of the primary goals of our work was to assess the reproducibility of the findings in [3] on continuous learning for Android malware detection. To ensure a direct and meaningful comparison, we adhered to using the same dataset in our study. However, if the reviewer believes incorporating additional datasets would strengthen our analysis and provide deeper insights, we would be happy to consider them and welcome any specific suggestions.
>
> - **The duplicates selection approach:** Both datasets used in our study rely on binary feature vectors, meaning the deduplication process does not encounter numerical precision issues. However, the reviewer is correct to point out that the feature extraction process may map two distinct raw samples to identical feature representations. We explicitly acknowledge this issue in Section 8 (Limitations & Discussions) of our paper.
> Deduplication at the input space level is a well-known challenge in malware detection and is not the primary focus of our study. Instead, our goal was to investigate how the presence of duplicates in the feature space affects the variability and reproducibility of reported results. From this perspective, we observe that duplicates introduce significant challenges in faithfully reproducing prior findings—an issue we elaborate on in Section 6.2. While one could argue that retaining duplicates in the training set is preferable, it remains essential to conduct variance analysis (particularly when duplicates are present) so that future studies can reliably reproduce reported results.
>
>
> - **Insights about the seed selection and reporting:** The reviewer suggests that publishing results with different random seeds may be beneficial but not essential. While this may hold for datasets with minimal variance due to stochasticity, this is not the case for our datasets—especially in the presence of duplicates, as we further discuss in Section 6.2. Without accounting for this variation when reporting results, reproducibility across studies becomes exceedingly difficult.
> Moreover, reporting results averaged over multiple seeds is a common practice in non-i.i.d. settings, such as domain generalization [1,2], which aligns with our study’s setup. This approach helps account for performance variance, ensuring a more robust and reliable evaluation.
>
>
> - **The key takeaway:** While our results show that XGBoost often outperforms various neural network architectures, we also observe that within the three neural network models we studied, additional model complexity can improve performance—particularly in continuous active learning with the Drebin dataset (Table 4). Therefore, we do not claim that simpler baseline models necessarily outperform more complex ones when properly trained. Instead, our findings indicate that XGBoost frequently outperforms neural network-based approaches for malware detection on the datasets we examined.
>
>
> **Response to Requested Changes**
>
> As mentioned earlier, we use the same two datasets as [3] to ensure a direct and meaningful comparison. However, our work extends [3] by exploring a broader range of experimental settings, including offline and continual learning as well as merged and holdout training strategies. If the reviewer has specific suggestions regarding additional datasets or experimental settings that could further strengthen our study, we would be happy to include them.
>
> We will incorporate any necessary revisions based on the reviewer’s feedback to further clarify insights from our results.
>
>
> **References**
>
> [1] Gulrajani, Ishaan, and David Lopez-Paz. "In Search of Lost Domain Generalization." ICLR, 2021.
> [2] Yu, Han, et al. "Rethinking the Evaluation Protocol of Domain Generalization." CVPR, 2024.
> [3] Chen, Yizheng, Zhoujie Ding, and David Wagner. "Continuous Learning for Android Malware Detection." 32nd USENIX Security Symposium (USENIX Security 23), 2023.

---

### Review · Reviewer_jAYY · 2025-02-22

**Summary Of Contributions:**

This submission argues that:

For android malware detection based on static features, factors such as data duplication, hyper-parameter tuning and model selection have sizable impacts on the benchmarking results.
Different methods/models can be preferred as these factors vary, posing reproducibility/replicability challenges to the field.

As a result, they present an evaluation pipeline with data deduplication, hyper-parameter tuning and model selection specified, which is suggested as a standardized framework for future studies.

**Audience:**

Yes

**Broader Impact Concerns:**

No significant concern is noted.

**Claims And Evidence:**

No

**Requested Changes:**

I think this submission identified several factors that could affect evaluations, but the supports for a more reliable/reproducible evaluation design were missing. This is critical.

I would suggest the followings:
1. Duplicated v.s. Deduplicated: I would like to see justifications regarding why deduplication (based on static features) is considered a better evaluation protocol or not. This is not clear to me based on the current submission.

2. Hyper-parameter tuning: I would like to see analysis/insights regarding how good the currently used hyper-parameter tuning is. I would definitely suggest delving into the candidates in Table 7 and expand in a smarter way. For example, one might argue that using "coordinate descent" or simulated annealing as heuristic search could greatly favor more complicated models. I do not see good justifications regarding random search being a more reliable/reproducible way of hyper-parameter tuning for benchmarking purposes.

**Strengths And Weaknesses:**

**Strength:**

The submission offers sufficient empirical supports demonstrating whether factors such as data duplication, hyper-parameter tuning and model selection can affect evaluation results, which justifies the need for making them explicit when benchmarking different models/methods.

**Weakness:**

1. The submission fails to justify whether/why their evaluation protocol is more reliable or more reproducible in many aspects. Specifically, the followings are not adequately addressed:
- Duplicated v.s. Deduplicated: They yield different benchmarking results, but why using deduplication (based on static features) is a more reliable protocol? Also "using duplicated datasets in an active learning setting is impractical" is in fact not well justified in continuous learning settings, even with static features. This is because there could be distribution shifts over time, causing correct annotations for "duplicate" samples to change.
- Hyper-parameter tuning: Why is random search with 200/100 sampling a reliable protocol? It could still be highly sensitive to the range of candidate values, especially for XGBoost/MLP/SCC/HCC as they have more hyper-parameters (larger search spaces). One particularly notable example is for HCC, where a shared set of learning rates (1e-3 ~ 7e-3) are used regardless whether the optimizer is Adam or SGD, which is a clear waste of computes for tuning as Adam and SGD typically require very different learning rates.

2. The scope of this submission is set to be quite narrow/specific, malware detection but (1) only for Android and (2) assuming static feature (and I believe only Drebin feature is considered as the two benchmarks are based on the same feature).

---

> ### Author Response · Authors · 2025-02-26
> **Response to Reviewer jAYY**
>
> We sincerely appreciate the reviewer's insightful suggestions and comments.
>
> **Response to Weaknesses**
>
>
> **1(a) Duplicated vs. Deduplicated Datasets:**
> One of the central premises of our work is to address the reproducibility of prior research [3] on continuous learning for Android malware detection using static features. From this perspective, we argue that deduplicated datasets provide a more reliable estimate of model performance on the test set compared to duplicated datasets. This is because performance variations in a few samples can significantly influence the results when duplicates are present.
>
> As we observe in these datasets, up to 90% of samples are duplicated in certain months with identical input features. This high redundancy introduces significant performance fluctuations due to stochastic variations in model predictions on a single instance. We elaborate on this issue in Section 6.2 of our paper.
>
> The reviewer is correct that, in some cases, correct annotations may change due to distribution shifts. However, our deduplication procedure for the continuous learning setup already accounts for this factor. As discussed in Section 4.1.2, in the continuous learning setup, we only deduplicate samples within the same evaluation period (i.e., each month). This means that the same sample may still appear in two or more distinct months, even after deduplication.
>
>
> **1(b) Hyperparameter Tuning:**
> We employ random search for hyperparameter tuning because it is the standard practice in the domain generalization literature, where training and test datasets do not adhere to the i.i.d. assumption [1,2]. The primary rationale behind this choice is to ensure a fair comparison across different methods. Unlike more exhaustive search strategies, random search prevents more complex models from gaining an unfair advantage simply due to a larger hyperparameter space. To maintain rigor, we set the number of trials sufficiently high to ensure that all the models we considered reached stable performance on the validation set.
>
> To ensure a fair and efficient evaluation, we allocated a hyperparameter search budget of 200 trials for offline learning and 100 trials for continuous learning. This decision was guided by empirical observations showing that model performance on the validation set consistently converged within these iterations. As further justification, we present supporting analysis in Appendix C.3 (Figures 7 and 8), demonstrating how performance stabilizes as hyperparameter trials increase.
>
> Regarding HCC, we adopt the same hyperparameter search space for the optimizer used in the original study [3].
>
>
>
> **2. Scope of the Study**
>
>
> We use the same feature sets as the original study [3]—Drebin and APIGraph—which are based on distinct feature representations. Moreover, these datasets were collected from different time periods, enhancing the generality of our analysis.
>
> Our work extends the original study in several key ways. For example, we conduct a separate analysis of offline and continuous learning settings and explore multiple evaluation protocols, including merged and holdout training. If the reviewer believes incorporating additional feature sets would further strengthen the study, we would be happy to consider them and welcome any specific suggestions.
>
>
>
>
> **Response to Requested Changes:**
>
>
> 1. Duplicated v.s. Deduplicated: Please refer to our response to Weakness 1(a) for a detailed discussion.
> 2. Hyperparameter tuning: Please see our response to Weakness 1(b).
>
>
> **References**
>
> [1] Gulrajani, Ishaan, and David Lopez-Paz. "In Search of Lost Domain Generalization." ICLR, 2021.
> [2] Yu, Han, et al. "Rethinking the Evaluation Protocol of Domain Generalization." CVPR, 2024.
> [3] Chen, Yizheng, Zhoujie Ding, and David Wagner. "Continuous Learning for Android Malware Detection." 32nd USENIX Security Symposium (USENIX Security 23), 2023.

---

### Review · Reviewer_ZvXQ · 2025-03-06

**Summary Of Contributions:**

The paper proposes a more rigorous methodology for the training and testing of machine learning models for malware detection using features from static analysis. The authors argue that the related work often contains inconsistencies in the experimental methodology which has a negative impact in reproducibility of results. The paper discusses some of these pitfalls and proposes solutions to address them. The paper also includes an experimental evaluation using different machine learning models and configurations, showing that algorithms like XGBoost often offer a very competitive performance when compared to more complex machine learning models.

**Audience:**

Yes

**Claims And Evidence:**

No

**Requested Changes:**

+ Analyze the problem of the variance in performance looking also at the ROC-AUC metric. Analyze if the problem is caused by the data imbalance and the lack of proper calibration of the classifier’s threshold.

+ Provide a more thorough analysis of the impact of duplicates and a justification of the use of one or another approach for removing duplicates.

+ Include experiments using malware from other platforms (e.g., EMBER or SLEIPNIR for Windows malware).

+ Revisit the model selection strategy in Section 5.3.

**Strengths And Weaknesses:**

Strengths:
+ The use of an adequate experimental methodology for training and evaluating machine learning models for malware detection is a very relevant problem that is often overlooked in many papers in the research literature. The paper analyzes four interesting aspects that has an impact on this: 1) duplicates; 2) model selection strategies; 3) accounting for variance in performance; and 4) delayed evaluation of models.

+ The paper is well written and organized and easy to follow. It also includes a nice coverage of the related work in malware analysis.

+ The authors provided the source code for reproducibility purposes.

Weaknesses:
+ The problem is relevant to the area of knowledge. However, the value of the contribution is not very clear: one the one hand other papers in the related work has treated some of these aspects. On the other side, in many cases, the methodology proposed is rather common for the training and testing of machine learning models in other application domains beyond malware. In my view some of the choices made by the authors are not well discussed and justified and can be arguable (see my comments below).

+ The authors mention that there is a high variance in the results, especially for some models like neural networks. However, in my opinion, the nature of this variance is not well discussed in the paper. Thus, on one side, the authors just considered metrics like the F1-score and the false positive and negative rates. However, other metrics like the ROC-AUC can provide a more global view of the performance of the model (regardless of the threshold used to adjust the sensitivity of the ML model) and, possibly, the ROC-AUC is a more stable metric in this case. On the other side, the datasets used present a notable data imbalance (90% benign applications vs 10% malware), which can also provoke instability during the training of the models. Then, the variance problem can also be perhaps caused by the lack of implementation of rebalancing techniques to mitigate data imbalance. Finally, the variance on the F1-score possibly points out a problem related to the calibration of the threshold of the ML models. In this sense, for example, adjusting the threshold to maximize the F1-score on the validation set could be a reasonable strategy to reduce the variance and obtain more stable results.

+ The removal of the duplicates for both the training and test sets is arguable. It would be good to know the reason why these duplicates exist: they can be the result of a sloppy data collection strategy, or they can reflect the frequency of analysis of different applications and malware samples. In the second case, removing the duplicates (especially in the test set) can provide a biased view of the classifier’s performance: for instance, an error on a very frequent type of malware has little impact on the performance when using deduplication, but from a practical standpoint, misclassifying a very frequent type of malware can have very negative consequences. Similarly, removing duplicates in the training set, can reduce the emphasis given to very common types of malware or benign applications, which again, can be detrimental from a practical perspective. On the other hand, removal of duplicates across training, validation, and test set helps to analyze the ability of the classifier to detect new types of malware, but it does not show its capabilities to classify existing malware (i.e., malware that has already been analyzed). In other words, deduplication across datasets changes the data distribution and can provide an unrealistic view of the classifier’s performance. I think all these aspects should be better discussed and analyzed in the paper.

+ In Section 5.3, for model selection, the authors use random search with a fixed budget of 200 searches per model on the validation set claiming that this provides a fair comparison. However, it is unclear to me why this provides a fair comparison. Algorithms, especially if the search space is different. It really depends on what is the design criteria: training time efficiency vs performance. In this sense, for example, this can lead to poor results in more powerful models with more hyperparameters. Apart from this, random search is a very basic approach for hyperparameter search. It would be more beneficial to use more advanced search strategies.

+ In the experiments the authors just considered Android malware. Given the scope of the paper, I think it would benefit from including other types of platforms (e.g. Windows malware).

---

> ### Author Response · Authors · 2025-03-20
> **Response to Reviewer ZvXQ**
>
> We appreciate the reviewer's insightful suggestions and comments.
>
> **Response to Weaknesses**
>
> **1. Contribution:** While our methodology aligns with standard machine learning practices, our study introduces several novel contributions to Android malware classification. First, we examine the impact of duplicate samples on reproducibility, highlighting an often-overlooked issue. Second, we explicitly incorporate the temporal nature of data splits into our evaluation, making our study the first to introduce and analyze both merged and holdout evaluation settings for malware detection. Finally, we highlight the impact of proper hyperparameter tuning in active learning, demonstrating that XGBoost significantly outperforms other models, contrary to previously reported findings. These contributions fill important gaps in the field and provide a more rigorous foundation for evaluating Android malware detection models.
> **2. Variance Analysis:** We use the macro-F1 score as the evaluation metric because it is the de facto standard in Android malware classification and has been widely adopted in prior studies that address concept drift and employ temporal data splits [1, 2, 3, 4]. The macro-F1 score provides a balanced performance assessment across all classes, making it particularly suitable for scenarios with class imbalance, which is common in real-world malware detection tasks. In contrast, while AUC is a popular metric in many classification problems, it is less suitable in our setting due to the combined challenges of concept drift and data imbalance in the test set and may provide an overly optimistic assessment.
>
> We incorporated data balancing as part of the hyperparameter tuning process for all baseline models (please refer to Appendix C, Table 7).
>
> We have included a PDF (see the supplementary material) with results on threshold optimization to maximize the F1-score on the validation set and its impact on variance in test set performance for malware detection. While tuning improves performance, it does not eliminate performance variance.
>
> **3. Impact of duplicates:** We acknowledge the reviewer’s concern that removing duplicates can bias performance assessment and agree that its necessity depends on context. We do not claim it is always the best approach but emphasize that variance in performance must be appropriately accounted for (Section 8). Our findings (Section 6.2) show that deduplication improves reproducibility and reveals key performance trends, leading to a more realistic model evaluation.
>
> **4. Model selection:** We use random search for model selection, following standard practice in domain generalization where training and test sets violate the i.i.d. assumption [1,2]. This ensures a fair comparison across methods, preventing more complex models from benefiting unfairly from a larger hyperparameter space. To maintain rigor, we allocate 200 trials for offline learning and 100 trials for continuous learning, as empirical observations show that performance on the validation set consistently stabilizes within these iterations. Additional analysis in Appendix C.3 (Figures 7 and 8) further supports this choice by illustrating how model performance in the validation set converges as the number of hyperparameter trials increases.
>
> **5.Extension:** While our study focuses on Android malware, these features have been widely adopted in prior work on Android malware detection [1, 2, 3, 4, 7]. However, unlike previous studies, we explicitly address reproducibility challenges and emphasize the importance of proper experimental settings. While we believe our findings may generalize to other domains, such as Windows malware detection, an initial analysis of Windows datasets (e.g., EMBER, BODMAS) suggests that feature-space duplicates are far less prevalent. Consequently, the impact of feature-space duplicates on reproducibility and evaluation biases in Windows malware detection is likely minimal compared to Android domain.
> **Response to Requested Changes:** Addressed above.
>
> References:
> [1] Jordaney, Roberto, et al. "Transcend: Detecting concept drift in malware classification models." 26th USENIX security. 2017.
> [2] Pendlebury, Feargus, et al. "TESSERACT: Eliminating experimental bias in malware classification across space and time." USENIX Security. 2019.
> [3] Barbero, Federico, et al. "Transcending transcend: Revisiting malware classification in the presence of concept drift." IEEE Symposium on Security and Privacy (SP). 2022.
> [4] Chen, Yizheng, Zhoujie Ding, and David Wagner. "Continuous learning for android malware detection." USENIX Security 23. 2023.
> [5] Gulrajani, Ishaan, and David Lopez-Paz. "In Search of Lost Domain Generalization." ICLR, 2021.
> [6] Yu, Han, et al. "Rethinking the Evaluation Protocol of Domain Generalization." CVPR, 2024.
> [7] Zhang, Xiaohan, et al. "Enhancing state-of-the-art classifiers with api semantics to detect evolved android malware." ACM SIGSAC. 2020.

---

### Decision · Action_Editor_Lckc · 2025-04-14

**Recommendation:** Reject

**Comment:**

After carefully considering the reviewers' arguments and conducting my own evaluation of the paper, I think the paper is not suitable for TMLR. The paper is targeting the wrong audience, which likely explains why the reviewers noted a lack of novelty.

The paper emphasizes the importance of careful data preparation and fair comparisons—points that are already well-understood within the TMLR community. From a dataset perspective, the only contribution is the removal of duplicates, and no new algorithms or conceptual insights are presented.

The value of this work lies elsewhere. It is more likely to benefit researchers in communities such as USENIX, ACM SIGSAC, IEEE Security & Privacy, and NDSS, where the original papers using this data were published, and some experimental practices may not have followed the best standards. While I acknowledge that these venues are highly competitive, TMLR’s primary mission is to advance machine learning research. Unfortunately, this paper does not meet that criterion.

**Audience:**

This paper would be interesting for the TMLR audience that also works in security, which is not small, however, not a core demographic.

**Claims And Evidence:**

The paper focuses on the detection of malware on Android devices. The authors focused on the inconsistency of the databases, needing to reduce the duplicates and carry out proper experimentation. This set-up is well received by the reviewers.